# Does the Expression and Epigenetics of Genes Involved in Monogenic Forms of Parkinson’s Disease Influence Sporadic Forms?

**DOI:** 10.3390/genes13030479

**Published:** 2022-03-08

**Authors:** Aymeric Lanore, Suzanne Lesage, Louise-Laure Mariani, Poornima Jayadev Menon, Philippe Ravassard, Helene Cheval, Olga Corti, Alexis Brice, Jean-Christophe Corvol

**Affiliations:** 1Institut du Cerveau–Paris Brain Institute–ICM, Inserm, CNRS, Sorbonne Université, 75013 Paris, France; suzanne.lesage@upmc.fr (S.L.); louise-laure.mariani@aphp.fr (L.-L.M.); poornima.menon@icm-institute.org (P.J.M.); philippe.ravassard@icm-institute.org (P.R.); helene.cheval@icm-institute.org (H.C.); olga.corti@icm-institute.org (O.C.); alexis.brice@icm-institute.org (A.B.); jean-christophe.corvol@aphp.fr (J.-C.C.); 2Assistance Publique Hôpitaux de Paris, Department of Neurology, CIC Neurosciences, Hôpital Pitié-Salpêtrière, 75013 Paris, France; 3Assistance Publique Hôpitaux de Paris, Department of Genetics, Hôpital Pitié-Salpêtrière, 75013 Paris, France

**Keywords:** Parkinson’s disease, Parkinson’s and related diseases, epigenetic, neurodegeneration, DNA methylation, histone modification, genetic, RNA-based gene regulation

## Abstract

Parkinson’s disease (PD) is a disorder characterized by a triad of motor symptoms (akinesia, rigidity, resting tremor) related to loss of dopaminergic neurons mainly in the *Substantia nigra pars compacta*. Diagnosis is often made after a substantial loss of neurons has already occurred, and while dopamine replacement therapies improve symptoms, they do not modify the course of the disease. Although some biological mechanisms involved in the disease have been identified, such as oxidative stress and accumulation of misfolded proteins, they do not explain entirely PD pathophysiology, and a need for a better understanding remains. Neurodegenerative diseases, including PD, appear to be the result of complex interactions between genetic and environmental factors. The latter can alter gene expression by causing epigenetic changes, such as DNA methylation, post-translational modification of histones and non-coding RNAs. Regulation of genes responsible for monogenic forms of PD may be involved in sporadic PD. This review will focus on the epigenetic mechanisms regulating their expression, since these are the genes for which we currently have the most information available. Despite technical challenges, epigenetic epidemiology offers new insights on revealing altered biological pathways and identifying predictive biomarkers for the onset and progression of PD.

## 1. Introduction

Parkinson’s disease (PD) is a neurodegenerative disease, characterized by progressive degeneration of the dopaminergic neurons of the *Substantia nigra pars compacta*. Its pathology is multifactorial; influenced by both environmental and genetic determinants [1]. Several pathogenic mutations have been linked to autosomal dominant or recessive forms of PD. The discovery of these genes allowed a new insight into the pathophysiology of this disorder [2]. Pathological hallmarks of PD include the presence of cytoplasmic inclusions, called Lewy bodies (LB), mainly composed of aggregated αsynuclein (α-Syn) [3]. Multiplication and point mutations of *SNCA* encoding α-Syn are now recognized to cause autosomal dominant PD and they are suspected of promoting α-Syn aggregation. The *PINK1*, *PARKIN* and *DJ1* genes encode for proteins required by mitochondria, which are essential components of neurons for ATP synthesis, calcium storage, lipid metabolism and neuronal survival [2,4]. The fact that mutations in these genes lead to PD is a strong argument that mitochondrial dysfunction is involved in the pathophysiology of PD.

Genetic mutations account for only 10% of patients with PD, and therefore environmental exposure seems to play a major role in PD [2]. Epigenetic modulation of gene expression by environmental factors is increasingly studied. Epigenetic regulation involves different mechanisms such as modification of the histones of chromatin and DNA methylation changes [5]. Chromatin is a dynamic scaffold and modification of its main components, the histones, can modulate gene expression [6]. The effect of histone modification is mediated either by directly affecting the structure of chromatin, by disrupting the binding of proteins that associate with chromatin or by attracting certain effector proteins to chromatin [5]. Histone acetylation decreases the compression of chromatin and promotes gene transcription. Methylation of histone H3 lysine 4 (H3K4), H3K36 and H3K79 are marks of transcriptional activation, whereas methylation of H3K9, H3K27 and H4K20 are repressive modifications of transcription, involving the recruitment of methylating enzymes and HP1 to the gene promoter [7].

Direct DNA methylation is also a key epigenetic mechanism regulating gene expression. It is a reversible modification of DNA, which consists of the addition of a methyl group to the fifth carbon position of a cytosine, converting it to 5-methylcytosine (5mC). The transfer of a methyl group is carried out by DNA methyltransferase (DNMT) enzymes [8]. This epigenetic mark is frequently found within a 5’-Cytosine-phosphate-Guanosine sequence, called a CpG site [9]. DNA methylation is not homogeneously distributed in the genome, CpG sites are clustered in sequences called CpG islands (CGI). Methylation of promoter-associated CGIs can impair transcription factor binding or recruit repressive binding proteins, thus reduce gene expression [10]. Cytosine methylation is mediated by DNMTs, which can be classified as de novo (DNMT3A and DNMT3B) and maintenance (DNMT1) [11].

Epigenetic regulation is also closely linked to non-coding RNAs. Non-coding RNAs are classified according to their size, with small RNAs less than 200 nucleotides long and long non coding (LncRNAs) longer than 200 nucleotides [12]. Among the small non-coding RNAs, microRNAs (miRNAs) are the most studied. Mature miRNAs bind to complementary sequences of the target messenger RNA (mRNA), often in the 3′ untranslated region (3’UTR), and can increase mRNA degradation but also inhibit translation without reducing mRNA expression [13,14]. The regulation of gene expression by Long non-coding RNAs (LncRNAs) is not yet fully understood. However, they appear to be important genomic regulators, from the epigenetic to the post-translational level [15].

Epigenetic mechanisms influencing the development of sporadic PD have yet to be identified. Genes involved in monogenic forms of PD could be over- or under-expressed in the sporadic form compared to the general population without PD (controls). In the context of altered expression of these genes in sporadic forms, it can be assumed that epigenetic mechanisms may be involved in this dysregulation (Table 1).

We will review here the differences in the expression of these genes between sporadic human PD and controls. We will then consider whether histone modification, DNA methylation but also miRNA expression could account for the difference in expression. Finally, we will discuss whether the observed changes are epiphenomena or are an integral part of the pathophysiology of the disease. 

## 2. α Synuclein

### 2.1. Function and Subcellular Distribution of α-Syn in PD

α-Syn, encoded by SNCA, is a small protein expressed abundantly in neurons of the central nervous system and located mainly at the presynaptic level [42]. In addition to the synaptic localization, the protein has been detected in the nucleus, explaining the name “synuclein” [43,44]. α-Syn is predominantly a soluble and highly mobile protein and since its molecular weight is less than the nuclear pore cutoff (∼40 kDa), α-Syn can enter the nucleus by simple diffusion [44]. 

In PD, α-Syn monomers assemble into insoluble β-sheets-rich fibrils that together compose Lewy bodies (LB) [42]. It is assumed that the pathogenicity of α-Syn is caused by its accumulation and oligomerization preceding the formation of aggregates [45]. Post-mortem brain analysis of subjects without synucleopathy revealed that physiological α-Syn expression is lower in brain regions not subjected to LB accumulation [46].

The N-terminal domain of α-Syn allows its association with anionic phospholipids, preferentially binding to small vesicles [47,48]. It has been shown that α-Syn is involved in the regulation of synaptic vesicle trafficking and neurotransmitter release [42]. The association of α-Syn with lipids seems essential to allow the oligomers of this protein to disrupt the membranes and, consequently, to induce a dysfunction of the vesicular systems [49]. Several studies have reported that *SNCA* point mutations located in the N-terminal region of α-Syn p.A30P, p.A53T and G51D induce an increase in nuclear localization of α-Syn compared to the wild-type protein [42]. This data suggests that the N terminal region is functionally involved in the subcellular distribution of α-Syn. Other mechanisms proposed to be involved in its subcellular distribution include interactions with nuclear or cytoplasmic proteins, e.g., TRIM28 and oxidative stress [18,24,25]. 

The nuclear function of α-Syn is undetermined, but the presence of α-Syn in the nucleus seems to promote neurotoxicity, whereas cytoplasmic sequestration is protective in both cell culture and Drosophila [50]. This finding has also been supported by other studies involving SIRT2 inhibition [51].

### 2.2. Overexpression of α-Syn in PD

SNCA associated PD is characterized by duplications or triplications of the SNCA locus with the number of SNCA alleles correlating with the amount of α-Syn overexpression and also the severity of the clinical phenotype [52,53]. 

Correspondingly, the development of sporadic PD might be associated with increased SNCA expression or impairment of protein clearance mechanisms. However, studies on *SNCA* mRNA in brain regions of sporadic PD patients and controls are discordant, with some studies revealing even a decrease in *SNCA* expression in the *Substantia nigra* of PD patients [54]. A limitation of these studies is that the brain examination is performed at a late stage of the pathology, and due to neurodegeneration, the *SNCA* mRNA level might reflect expression in the remaining cells but not the expression in the affected neurons [54]. A study of surviving dopaminergic neurons laser-captured from the substantia nigra of post-mortem brains revealed an increase in *SNCA* mRNA in PD subjects compared to controls [16]. A splice variant of the SNCA gene, SNCA-126, has also been shown to be overexpressed in the *Substantia nigra* of PD patients [55]. In a study, the level of α-Syn protein was also modestly increased in the *Substantia nigra* of PD patients compared to controls [56]. 

Moreover, single nucleotide polymorphisms (SNPs) of *SNCA* have been identified in genome-wide associations studies (GWAS) as a risk factor that increases susceptibility to developing sporadic PD [57]. Among these variants a singular SNP, rs356168, in a non-coding distal enhancer element of *SNCA* leads to an increase in expression of α-Syn [58]. 

### 2.3. Epigenetic Regulation of SNCA Expression in Sporadic PD

DNA methylation of *SNCA* was reported to modulate its expression [17]. Hypomethylation in intron 1 of *SNCA* was observed in multiple brain regions of PD patients and was associated with increased *SNCA* expression in vitro [17,18]. Among the regulators of DNA methylation, DNMT1 appears to play an important role in *SNCA* expression. DNMT1 is mainly located in the nucleus of neurons and α-Syn aggregation leads to cytoplasmic sequestration of DNMT1 in animal models and patient brains [59] (Figure 1). This mechanism might explain the DNA hypomethylation at the *SNCA* gene and increased *SNCA* transcription. Hypomethylation of the *SNCA* promoter and increased *SNCA* expression in methamphetamine-exposed rats appears to be mediated by decreased occupancy of DNMT1 in the *SNCA* promoter region [60]. These observations were further associated with decreased nuclear localization of DNMT1 [60]. However, it is uncertain whether the observed hypomethylation is the response to cytoplasmic sequestration of DNMT1 or whether this mechanism is involved in the pathogenesis of the disease.

Hypomethylation of *SNCA* intron 1 has also been found in peripheral tissues of PD patients such as blood samples, peripheral blood mononuclear cells (PBMCs) and leukocytes [19,20,21]. The study of white blood cell *SNCA* methylation in healthy patients revealed a decrease in *SNCA* methylation with age [61]. Among healthy individuals, women had higher methylation of SNCA than men, which may contribute to the lower incidence of sporadic PD in women [61,62]. Furthermore, SNCA methylation in peripheral blood of sporadic PD patients was increased with higher doses of L-dopa and concordantly, L-dopa induced SNCA intron 1 methylation was observed in cultured mononuclear cells from PD patients [61]. These interesting findings highlight how environmental factors are correlated with epigenetic modifications.

An analysis of SNCA histone architecture in post-mortem midbrain samples found that three histone modifications, H3K4 trimethylation (H3K4me3), H3K27 acetylation (H3K27ac) and H3K27me3, were preferentially enriched in SNCA regulatory regions [63] (Figure 2). H3K27ac and H3K4me3 promote transcription and show a peak around the transcription start site [63]. Concordantly, a H3K27ac rich sequence was previously identified within the *SNCA* locus [64]. The involvement of histone modification in *SNCA* expression was first reported in a patient heterozygous for the SNCA p.A53T mutation. The repression of the mutated allele in this subject was not due to DNA methylation but due to histone deacetylation. The use of histone deacetylase (HDAC) inhibitor in cell models reactivated the mutated allele expression [65]. Epidemiological studies have shown that β-2 adrenergic receptor antagonists increase the risk of developing PD. Acetylation of H3K27 *SNCA* histones was proposed as the possible mechanism by which Adrenergic β2 receptor antagonists, potentially via inhibition of the β2-adrenergic receptor pathway, leads to accumulation of α-Syn [66]. However, the analysis of post-mortem midbrain samples found no significant difference for H3K27ac between PD subjects and controls [63]. While the histone mark H3K4me3 was enriched at the SNCA promoter in post-mortem brain samples from PD patients compared to controls. Furthermore, higher levels of H3K4me3 correlated with higher levels of α-Syn [63].

In addition, a-Syn itself is thought to play a role in regulating transcription through histone modification. It has been reported that nuclear α-Syn leads to transcriptional repression of PPARGC1A, encoding PGC-1α, potentially through decreased levels of histone acetylation. PGC-1α is a primary mitochondrial transcription factor involved in the regulation of mitochondrial biogenesis and oxidative metabolism [67,68]. It has been shown in mice that repression of PGC1-a by the PARIS protein leads to progressive loss of dopaminergic neurons [69].

### 2.4. Non-Coding RNAs Regulating *SNCA* Transcription

Translation of *SNCA* mRNA in the cytoplasm is regulated by specific miRNAs. It has been found that miR-7 and miR-153 binds directly to the 3′-UTR of *SNCA* mRNA, destabilizing the mRNA and significantly reducing its levels [70,71]. Neurotoxin 1-Methyl-4-Phenyl-Pyridinium (MPP+) induced decrease in miR-7 expression possibly contributes to increase *SNCA* expression in vitro and in mice [70]. In contrast, overexpression of miR-7 or miR-153 in cortical neurons by viral transduction showed a protective effect against MPP+ toxicity [72]. In vitro, miR-7 accelerates the clearance of α-Syn; an effect that seems to be mediated by its activation of autophagy [73]. 

Two other interesting mi-RNAs with multiple potential associations in the pathogenesis of PD are miR-34b/c. Reduced expression of these miRNAs were reported in the amygdala, *Substantia nigra*, frontal cortex and cerebellum of PD patients compared to controls [22]. A decrease in these miRNAs was also observed in the putamen of PD patients [74]. Depletion of miR-34b/c in differentiated SH-SY5Y neurons resulted in reduced cell viability, mitochondrial dysfunction and oxidative stress, and a slight decrease in DJ-1 and Parkin expression [22]. On the other hand, a study found that inhibition of miR-34b/c which targets the 3’-UTR of *SNCA* mRNA is associated with increased expression of mRNA *SNCA* and protein expression [75]. Alternative polyadenylation can lead to different 3′UTR lengths and at least five different 3′UTR lengths of SNCA transcripts have been reported [76]. *SNCA* transcript with longer 3′UTRs may promote protein accumulation and mitochondrial localization [76]. A study using luciferase-SNCA full length 3′UTR reporter vector reported that a miR-34b-mimic induced translation of the long 3’UTR SNCA transcript [76]. In addition, it has been shown that very low frequency magnetic fields decrease the expression of miR-34b/c in vitro [77]. This modulation could be attributed to CpG hyper-methylation within the miR-34b/c promoter observed with exposure to these fields [77]. 

Other non-coding RNAs might participate in the regulation of *SNCA* expression such as RP11–115D19.1. The expression levels of RP11-115D19.1-003 in the brains of healthy donors and PD patients were strongly and positively correlated with those of *SNCA* [78]. Knockdown of this LncRNA in a cell model led to an increase in *SNCA* expression, suggesting its repressive effect on *SNCA* expression [78]. 

### 2.5. Interplay between Epigenetic Mechanisms in the Regulation of SNCA

Different epigenetic mechanisms are involved in the regulation of α-Syn expression, however, the relative weight of each is not determined. Histone modifications such as H3K9me3, H3K27me3 and H4K20me1 cause local chromatin condensation and could lead to an easily reversible repression of gene expression [79,80]. DNA methylation appears to lead to long-term stable gene repression. As H3K4me3 is present in most of CGIs, regardless of whether the associated gene is actively transcribed or not, there appears to be a dependency between these two mechanisms [81]. H3K4me3 is particularly enriched in unmethylated CGIs, which may allow the maintenance of DNA hypomethylation and shape a chromatin environment that favors transcription [81]. This is consistent with the observations of DNA hypomethylation and H3K4me3 enrichment on the *SNCA* gene. However, variability in a-Syn expression was observed despite enrichment of H3K4me3 at the SNCA promoter, raising suspicion of complementary mechanisms. A multivariate analysis considering the level of different epigenetic mechanisms could be interesting to better discern the involvement of these mechanisms.

## 3. LRRK2

### 3.1. LRRK2 Protein Function and Localization 

*LRRK2* mutations induce the autosomal-dominant form of familial PD [82]. *LRRK2* encodes a serine/threonine kinase called dardarin, after the Basque word for tremor. The native protein appears to transit between a monomeric and dimeric form [83,84]. LRRK2 is involved in many cellular functions such as regulation of neurite growth and cytoskeletal dynamics, maintenance of lysosomal function, and synaptic vesicle endocytosis (SVE) [85]. After neurotransmission, the replenishment of synaptic vesicles with neurotransmitters is ensured by the SVE [86]. Some proteins involved in SVE such as synaptojanin 1 (SYNJ1), auxilin (DNAJC6), and endophilin A1 (SH3GL2) are also LRRK2 substrates [86]. Phosphorylation of these proteins by LRRK2 appears to result in SVE dysfunction [87]. The G2019S mutation is the most common LRRK2 mutation. This mutation located in the kinase domain results in an increased kinase activity of the protein and could be toxic by a gain-of-function mechanism [88] (Figure 3). In a Drosophila model of PD, G2019S *LRRK2* mutation suppresses the functions of let-7 miRNA and miR-184*, which regulate the translation of the E2F1/DP complex involved in cell cycle driving [89]. Moreover, LRRK2 also regulates gene transcription through the phosphorylation of HDAC3 by promoting histone deacetylation. In particular, LRRK2 leads to transcriptional repression of MEF2D, a gene associated with neuronal survival [90]. In addition to the G2019S mutation, at least six pathogenic *LRRK2* mutations have been identified as causative for PD that induce autosomal dominant PD [91]. The presence of non-coding variants in LRRK2 in sporadic PD, suggests that altered transcription of this gene is associated with the pathophysiology of sporadic PD [92].

It was reported that LRRK2 mRNA and protein levels differ between brain regions, with expression in target areas of the dopaminergic system, such as the striatum and frontal cortex, whereas neurons in the *Substantia nigra* showed very low mRNA and protein expression levels [93,94]. Consistent with this, immunohistochemical analysis of LRRK2 protein in *Substantia nigra* dopamine neuron bodies found no signal in either controls or PD cases [23]. 

### 3.2. Regulation of LRRK2 Transcription 

LRRK2 protein expression was higher in the frontal cortex and striatal neurons of sporadic PD patients compared to controls, in contrast to mRNA levels which did not vary between patients and controls [23]. In the frontal cortex of sporadic PD patients, an increase in LRRK2 expression was correlated with a decrease in miR-205. miR-205 is able to bind to the 3’UTR of *LRRK2* mRNA and suppress its expression [23]. In neuronal cultures expressing the *LRRK2* R1441G mutation, overexpression of miR-205 protected them from neurite growth defects [23]. The mechanism of miR-205 depletion in PD patients is, however, undetermined. In cancer, epigenetic modifications such as histone modification and DNA methylation, but also microenvironmental changes such as hypoxia, inflammation, and cytokines, contribute to miR-205 dysregulation [95]. It has been suggested that miRNA expression can be regulated by hypoxia in a tissue-specific manner [96]. MiR-205 was found to be induced by hypoxia in cervical and lung cancer cells, potentially through suppression of the apoptosis-stimulating protein p53-2 [97]. Furthermore, miR-205 expression was found upregulated in thymic epithelial cells following inflammatory responses where it helps to preserve the maturation of T cells in response to inflammation [98]. 

The transcription factor Sp1 promotes LRRK2 expression [99]. However, it has been shown that Sp1 induction also activates miR-205 expression [95], which may be a feedback mechanism for LRRK2 overexpression. 

LncRNAs also appear to play a role in the regulation of LRRK2. The LncRNA HOTAIR has been shown to increase *LRRK2* mRNA stability and increase its expression. HOTAIR is upregulated in neurons of MPP+-induced PD mice while its knockdown provides protection against MPP+-induced neuronal apoptosis [100].

### 3.3. LRRK2 in Immune Cells 

*LRRK2* is also expressed in immune cells (lymphocytes, monocytes, neutrophils and also microglia) where its expression is tightly regulated by immune stimulation, implicating its potential role as a regulator of immune responses [83,101]. *LRRK2* expression is higher in lymphocytes and inflammatory monocytes of late-onset PD patients compared to age-matched individuals, suggesting the role of inflammation in the development of PD [24]. This could explain why hypomethylation of the *LRRK2* gene is observed in leukocytes from PD patients [21]. *LRRK2* is also expressed in primary microglia from adult mice and it is upregulated upon IFN-y stimulation or lipopolysaccharide (LPS) treatment [102]. An iPSC study revealed that basal *LRRK2* mRNA expression was lower in sporadic PD microglia, and after treatment of the cells with LPS, sporadic PD microglia had a significantly lower amount of LRRK2 protein than control microglia [103]. However, the mechanisms linking LRRK2 downregulation to microglia dysfunction in PD remains to be elucidated.

## 4. Dysregulation of Genes Involved in Recessive Forms of PD

Genes involved in recessive forms of PD such as *PRKN*, *PINK1* and *DJ-1* are essential for physiological mitochondrial function.

### 4.1. Expression Profile of *PRKN* and Its Regulation

*PRKN* encodes Parkin, an E3 ubiquitin ligase. Parkin ubiquitinates various proteins, thereby promoting their proteasomal degradation. One of its roles is to control mitochondrial biogenesis, notably mediated by its influence on PGC1-α by ubiquitinating its repressor, PARIS [104]. In addition, Parkin is involved in cell survival signaling pathways [105].

With a genomic sequence above 1.38 Mb, the *PRKN* gene is the second largest in the human genome widely expressed in the brain [106]. More than 170 *PRKN* mutations have been associated with PD, including point mutations and genomic rearrangements [70]. In vitro, it has been reported that some mutations in *PRKN* result in a loss of the ubiquitin–protein ligase activity of Parkin [107]. A deletion in the promoter regions of *PRKN* resulting in the absence of the *PRKN* mRNA transcript has been associated with an early form of PD [108]. Parkin haploinsufficiency has also been identified as a risk factor for familial PD with a tendency towards an earlier age of onset [109]. Inactivation of *PRKN* in mice resulted in motor and cognitive deficits [110]. Overexpression of parkin or restoration of its activity leads to a protective effect against neurodegeneration in cell culture and in animal models [111,112,113]. These observations suggest that reduced expression of *PRKN* might confer a risk for developing PD. Post-translational modifications of Parkin induced by oxidative and nitrosative stress (sulfonation and S-nitrosylation) are increased in the brain of PD patients. It has been shown that these changes lead to a disruption of the E3 ligase activity of parkin and a dysfunction of the ubiquitin–proteasome system [114,115].

In addition, sulfonation and S-nitrosylation of Parkin alter the solubility of the protein, promoting its intracellular aggregation [115,116]. The decrease in the availability of soluble Parkin through aggregation could be involved in the pathophysiology of the disease.

The study of *PRKN* expression is complicated by the presence of different *PRKN* mRNAs due to alternative splicing of the gene [117]. To date, 26 *PRKN* transcripts have been identified, corresponding to 21 different alternative splice variants [118]. The pattern of *PRKN* expression differs between tissues and cells, with distinct splice variants in human brain regions and leukocytes [117]. Alternative splicing of non-coding sequences can influence the stability, translational activity and subcellular localization of transcripts; [119] while alternative splicing of coding sequences can generate protein isoforms with different biological properties. Alternative splicing could be regulated by LncRNA [120].

Distinct Parkin isoforms have been found to be differentially expressed in specific regions of the rat brain [121] and several isoforms of Parkin have been identified in human blood cells [122]. These observations suggest that the profile of Parkin isoforms may differ between human cells and tissues [118]. Parkin isoforms may have different subcellular locations as well as different functions. Recently it has been shown that intranuclear Parkin could change the transcriptional activity of genes involved in regulating multiple metabolic pathways through interaction with transcription factors [123]. 

### 4.2. Role of Parkin and Its Epigenetic Regulation in Sporadic PD

Even though reduction of *PRKN* is potentially involved in the pathophysiology of monogenetic forms of PD, *PRKN* expression does not appear to be decreased in the sporadic form. TV3 and TV12 variants of *PRKN*, resulting from alternative splicing, were overexpressed in the frontal cortex of sporadic PD compared to controls [25], suggesting a change in the expression pattern of *PRKN*. Another study reported that 3, 7, and 11 *PRKN* transcripts were overexpressed in the striatum and cerebellar cortex of PD patients compared to controls [124].

Studies on small cohorts of PD patients revealed no differences in PRKN methylation levels in blood and brain [26,27]. A recent study of blood samples from 91 early-onset PD patients showed hypomethylation of the PRKN promoter in this group compared to healthy controls [28]. However, this hypomethylation may not explain a reduction in PRKN expression and could rather correspond to a compensatory mechanism. Data comparing samples of PD patients and healthy subjects revealed a reduction in miR-181a in the serum [29] and a reduction in miR-218 in the brain of PD patients [30]. In vitro, overexpression of miR-181a and miR-218 each, induced a reduction in PRKN mRNA [125,126]. Among the miRNAs increased in brain or plasma samples from PD patients, some showed in silico binding sequences to PRKN 3’-UTR mRNA. It was confirmed that a selected miRNA, miR-103a-3p [31], directly regulates PRKN mRNA translation leading to a downregulation of Parkin protein level [127]. Furthermore miR-103a-3p inhibition improved mitophagy and had neuroprotective effects in PD models in vitro and in vivo [127]. 

### 4.3. PINK1 Expression in PD Patient Brains

The *PINK1* gene encodes a PTEN-induced serine kinase located in mitochondria that protects against mitochondrial dysfunction and regulates the mitochondrial fission/fusion mechanism. PINK1 is imported into the mitochondria and rapidly cleared by the proteasome. However, stress factors can lead PINK1 to accumulate in the outer mitochondrial membrane. PINK1 will then homodimerize, resulting in autophosphorylation, which promotes kinase activation and facilitates binding to its substrates Parkin and ubiquitin [104]. Parkin activation forms ubiquitin chains and this mechanism allows more Parkin to be recruited to the mitochondria, amplifying the damage detection signal. By ubiquitinating various proteins in the outer mitochondrial membrane, Parkin then initiates mitophagy (Figure 4). Alteration of this pathway can lead to the accumulation of dysfunctional mitochondria which may contribute to the loss of dopaminergic cells [104]. 

Most *PINK1* mutations are point mutations, small insertions or deletions, however, deletions of the entire gene and large complex rearrangements have also been reported [107]. By affecting several mitochondrial phases, including fission, fusion and mitophagy, *PINK1* mutations could lead to respiratory chain dysfunction and impaired ATP production [128]. 

The full-length form of PINK1 (FL-PINK1) imported into mitochondria undergoes proteolysis to produce ∆1-PINK1 which is then relegated to the cytosol and interacts with Parkin [129]. Binding of ∆1-PINK1 to Parkin impairs the recruitment of Parkin to mitochondria and represses mitophagy [129]. In the brains of PD patients, level of PINK1 mRNA were reported to not differ significantly from those of healthy subjects [33]. Accordingly, a recent study found no difference in *PINK1* methylation in the brains of PD patients compared to controls [34]. 

NFκB levels are elevated in dopaminergic neurons of Parkinson’s disease patients, reflecting an apoptotic and inflammatory state. In vitro PINK1 expression appears to be positively regulated by NFκB. An increase in ∆1-PINK1 level has been reported in the *Substantia nigra* of post-mortem PD patient brains compared to controls [32]. *PINK1* translation appears to be critical for the accumulation of the protein during mitochondrial damage. It has been reported that miR-27a/b represses *PINK1* expression by direct binding to the 3′UTR of its mRNA. PINK1 accumulation upon mitochondrial damage was regulated by miR-27a/b expression levels. The latter inhibits *PINK1* translation suppressing autophagic clearance of damaged mitochondria. Furthermore, miR-27a/b expression is increased under chronic mitophagic flux, suggesting a negative feedback regulation between PINK1-mediated mitophagy and these miRNAs [130]. Studies have reported a decrease in miR-27a in PBMCs as well as in cerebrospinal fluid in early-onset PD compared to controls [35,36]. The implication of this observation in the pathophysiology is still unclear. 

### 4.4. Regulation of DJ-1 Expression

*DJ-1* is expressed in many tissues and cells, including neurons and glial cells [131]. DJ-1 has an antioxidant function, notably through the elimination of reactive oxygen species (ROS) [132]. In the presence of oxidative stress, cytoplasmic DJ-1 is translocated to the outer mitochondrial membrane and is thought to play a role in neuroprotection [133]. Different isoforms of *DJ-1* have been identified in brain tissue. Analysis of post-mortem brain samples revealed a decrease in *DJ-1* mRNA and protein, as well as the presence of extra-oxidized *DJ-1* isoforms in subjects with PD compared to controls [37]. Acidic isoforms of the *DJ-1* monomer were selectively accumulated in the brains of sporadic PD patients compared to controls [134]. Depletion of DJ-*1* in vitro increases the sensitivity of cells to oxidative stress [135]. However, no significant association was found between polymorphisms within the *DJ-1* gene promoter and the risk of PD [136].

At the epigenetic level, the DNA methylation of *DJ-1* promoter in leukocytes was not different between PD patients and controls [38]. One brain-enriched miRNA, miR-494, was reported to bind to the 3′ UTR of *DJ-1* mRNA and reduce its level [137]. After treating SH-SY5Y cells with MPP+, miR-494-3p expression was increased [138]. In addition, mice overexpressing miR-494 treated with MPTP showed decreased expression of *DJ-1* with exacerbated degeneration of DA neurons and worsened motor impairment [138]. 

By binding to the 3’UTR of *DJ-1* mRNAs, another miRNA, miR-4639-5p, represses DJ-1 translation. Overexpression of miR-4639-5p in MPP+-treated SH-SY5Y cells results in reduced DJ-1 protein and increased vulnerability to cellular stress [39]. miR-4639-5p expression was higher in the plasma of PD patients than in controls, suggesting its potential role in the pathophysiology of PD [39].

## 5. GBA a Risk Factor for PD

In PD, clearance of the autophagy–lysosome pathway has been shown to be inefficient, since pathological aSyn can potentially compromise several stages of this pathway [86]. Mutations in the *GBA* gene encoding the lysosomal enzyme, glucocerebrosidase (GCase) involved in the lysosomal storage disorder, Gaucher’s disease, have been identified as the most common PD risk factor, highlighting the key role of lysosomal dysfunction in PD [86]. Post-mortem brain analysis revealed a decrease in GCase protein level and activity with increased α-Syn levels in PD patients compared to controls [40]. However, this decrease in the GCase activity does not seem to correlate with a low level of *GBA* mRNA expression [40,41]. 

To date, whole genome methylation studies have not demonstrated differential methylations of the *GBA* gene [28]. It was shown that miR-127-5p decreases GCase activity and protein levels, this effect being indirectly mediated by a decrease in LIMP-2, a receptor involved in GCase trafficking from the endoplasmic reticulum to the lysosome [139]. In addition, miR-16-5p and miR-195-5p have been shown to increase *GBA* transcript and GCase protein levels. However, their pathophysiological mechanism is not elucidated [139]. Recently, another hypothesis has been explored, based on the observation that specific lncRNAs can limit miRNA activity by sequestration, thus upregulating the expression of target genes. In this context, transcribed pseudogenes, nonfunctional segments of DNA that resemble functional genes, are ideal candidates because they share miRNA binding sites with the transcripts of target genes. In vitro, it has been reported that miR-22-3p directly binds to *GBA* and its pseudogene *GBAP1*, thus downregulating these two genes by decreasing their mRNA levels. Overexpression of *GBAP1* 3′ UTR in cell lines resulted in sequestration of mir-22-3p, thereby increasing *GBA* mRNA and GCase levels [140]. It is possible that dysregulation of miR-22-3p or *GBAP1* may participate in the pathophysiology of PD. 

## 6. Discussion

The expression of genes involved in familial forms of PD seems to also be dysregulated in sporadic PD patients. However, it remains unclear whether the change in gene expression corresponds to a pathophysiological mechanism, a marker of degeneration or a protective effect. 

For *SNCA*, its overexpression seems to be a prerequisite for aSyn aggregation leading to Lewy body (LB) formation [45]. Hypomethylation of *SNCA* could be involved in its overexpression. A pathophysiological mechanism suggested is the sequestration of cytoplasmic DNMT1 by the aggregates leading to the dysregulation of aSyn homeostasis [59]. The decrease in the brain level of miR-34b/c could also participate in the overexpression of *SNCA* [75]. 

The increase in LRRK2 kinase activity mediated by the G2019S and other disease-causing mutations raises the question whether *LRRK2* is overexpressed in sporadic PD [88]. Increased expression of *LRRK2* was found in the brains of PD patients, without increased transcripts [23]. A step in the pathophysiology could be a decrease in miR-205 expression, but the causes of this dysregulation is not clear [23]. Furthermore, the regulation of *LRRK2* expression by immune stimulation in blood tissue suggests the implication of neuro-inflammatory mechanisms [101] and could represent an accessible epigenetic biomarker. 

Due to alternative splicing and multiple splice isoforms, it is more challenging to understand the relationship between Parkin expression and its pathophysiology in sporadic PD [117]. Epigenetic mechanisms could be involved in the modification of the *PRKN* expression profile. 

Mitochondrial damage could induce overexpression of PINK1 [130]. However, the role of increased *PINK1* expression is unclear. It could be either protective or participating in neuronal death. Inflammation in PD seems to be most intense at the beginning, just after clinical diagnosis, attenuating in later stages [141,142]. It has been reported that miR-27a is downregulated in macrophages after stimulation [143]. The decrease in miR-27a observed in early PD may be involved in the inflammation-induced upregulation of PINK1.

Decreased transcription and translation of *DJ-1* in sporadic forms of PD may also be involved in the pathophysiology of the disease [37]. Oxidative stress leads to the expression of miR-494-3p [144]. The induction of miR-4639-5p expression by oxidative stress remains to be explored. We can hypothesize that miRs could be sensors of oxidative stress and contribute to the cellular response by downregulating DJ1.

Decreased GCase protein levels and activity may also be involved in the development of sporadic PD [40]. 

MiRNA and lncRNA mechanisms may be involved in the regulation of *GBA* expression but their change in expression level in sporadic PD patients is not clear. The mechanisms regulating miRNA expression are not well elucidated. An interesting approach would be to explore DNA methylation patterns or histone modifications at the transcription sites of these miRNA-regulating genes involved in monogenic forms of PD. Furthermore, epigenetic modifications involve tissue-specific processes. The observation of these mechanisms in pathologically relevant cell types and access to these cells is complex.

Analysis of native brain tissue allows us to observe epigenetic changes that take place in the brains of PD subjects. However, these analyses are limited to the post-mortem brain and longitudinal studies cannot be performed to assess the dynamics of epigenetic mechanisms. The collection of more accessible peripheral tissues such as blood or CSF allows an analysis in living individuals over time. However, since the methylation profile is tissue-specific, observations in these tissues do not confirm identical modifications in neurons. Since LRRK2 expression seems to play a role in the inflammatory response, the study of the regulation of its expression in the blood immune cells of PD patients could be interesting. The use of animal models of PD for the analysis of epigenetic alterations presents some difficulties since epigenetic mechanisms are species-specific, and these models may not be representative of epigenetic alterations in humans. However, they can allow for the study of specific neuronal populations [145].

Human iPSC-derived neurons have been developed, which allows for the generation of specific neuron lineages such as dopaminergic neurons derived from PD patients [146,147]. However, the generation of these cells requires cellular reprogramming based on epigenetic modifications resulting in “epigenetic rejuvenation” [148] and may bias the study of epigenetic alterations associated with age in PD. More recent direct conversion techniques make it possible to obtain induced neurons (iNs) with the same epigenetic age as their original fibroblasts. This conversion leaves the most age-related epigenetic marks intact but nevertheless leads to a reorganization of large parts of the epigenome [149]. On the other hand, significant epigenetic changes are largely cell type specific and the value of maintaining epigenetic marks in adult fibroblasts is controversial.

Although these in vitro models have limitations, they offer the opportunity to directly study putative effects of epigenetic modifications on gene expression. Moreover, they allow the development of 3D brain organoids or spheroids (cerebral organoids) with a better reproduction of the cerebral environment. This enables for better understanding of the epigenetic modifications that take place in neurons and other cell lineages such as astrocytes and microglia, also suspected in the pathophysiology of PD [147]. In addition to providing an understanding of the pathophysiology, epigenetic modifications could also allow for the development of biomarkers for the diagnosis, prognosis and monitoring of PD [150]. The demonstration of the causality of epigenetic mechanisms in the onset or progression of the disease could allow for the emergence of new therapeutic targets.

Epigenomic identity may also be mediated by chromosome folding [151]. Recent studies reveal that the 3D organization of the genome correlates with epigenetic modifications and that these modifications predict the structure of chromatin [151,152]. Moreover, changes in the 3D architecture of chromosomes have been observed in cancer cells [153]. This new field of analysis could allow a better understanding of the involvement of epigenetic modifications in PD.

## Figures and Tables

**Figure 1 genes-13-00479-f001:**
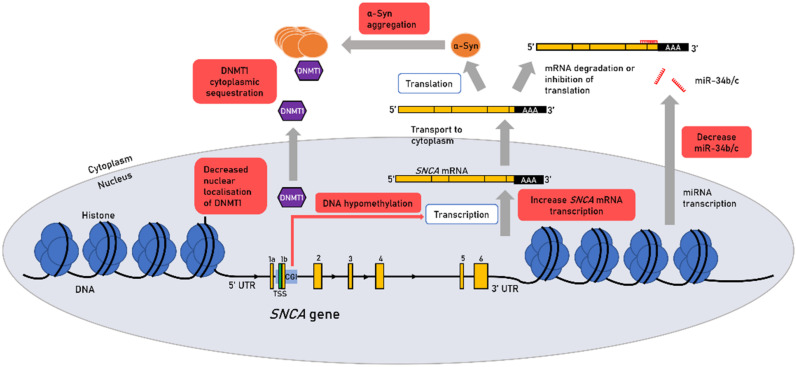
Epigenetic mechanisms and dysregulation of α synuclein. Abbreviations: 5′UTR: 5′ untranslated region; 3′UTR: 3′ untranslated region; 1–6: exon 1 to 6; CGI: CpG island; AAA: polyadenylation; DNMT1: DNA methyltransferase 1; mRNA: messenger RNA; miRNA: microRNA; TSS: transcription start site.

**Figure 2 genes-13-00479-f002:**
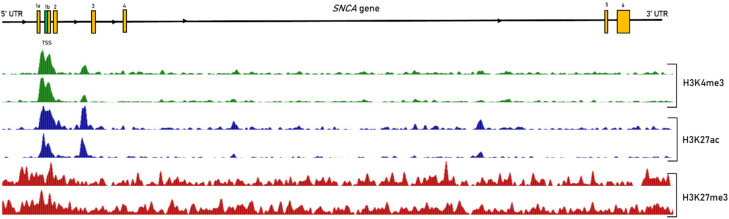
Distribution of histone modifications in the *SNCA* gene. Abbreviations: 5′UTR: 5′ untranslated region; 3′UTR: 3′ untranslated region; 1–6: exon 1 to 6; TSS: transcription start site. Histone modifications, H3K4me3 (green), H3K27ac (blue) and H3K27me3 (red), in the *SNCA* gene from *Substantia nigra* tissues of two healthy adult postmortem brain samples. Adapted from Roadmap Epigenomics Database.

**Figure 3 genes-13-00479-f003:**
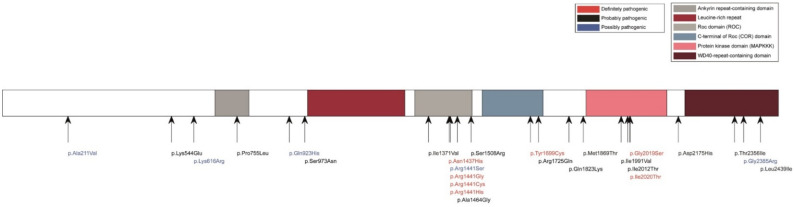
Current known variants in LRRK2 illustrated at the protein level (image obtained from MDS genes). The protein kinase domain of the LRRK2 protein where the G2019S mutation is located is visible in pink in the figure.

**Figure 4 genes-13-00479-f004:**
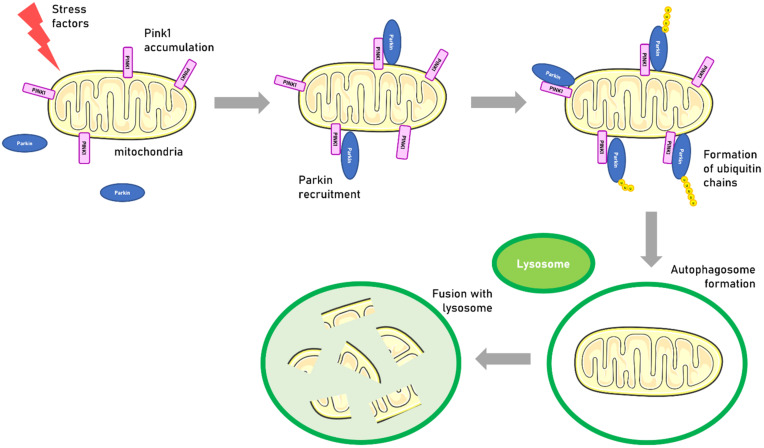
Scheme of mitophagy induced by PINK1 and Parkin. Abbreviation: u: ubiquitin. Under stress factors, PINK1 accumulates and recruits Parkin to the outer membrane of mitochondria. PINK1 and Parkin signaling allows autophagosome formation. Fusion of the autophagosome with the lysosome leads to mitophagy.

**Table 1 genes-13-00479-t001:** Expression profile and epigenetic changes observed for genes involved in monogenic forms of PD.

Studies	Tissues Analyzed	Proteins in Controls vs. sPD	mRNA in Controls vs. sPD	DNA Methylation in Controls vs. sPD	MiRNA Expression in Controls vs. sPD	Reference
SNCA						
Grundemann et al., 2008	Brain: DA neurons of SN	Increase	Increase	-	-	[16]
Jowaed et al., 2010/Matsumoto et al., 2010	Brain	-	-	Hypomethylation	-	[17,18]
Pihlstrom et al., 2015/Ai et al., 2014/Tan et al., 2014	Blood immune cells	-	-	Hypomethylation	-	[19,20,21]
Minones-Moyano et al., 2011	Brain	-	-	-	Decrease miR-34b/c	[22]
LRRK2						
Cho et al., 2013	Brain: frontal cortex/striatum	Increase	No difference	-	Decrease miR-205	[23]
Cook et al., 2017	Blood	Increase	-	-	-	[24]
Tan et al., 2014	Blood immune cells	-	-	Hypomethylation	-	[21]
PRKN						
Beyer et al., 2008	Brain	-	Increase in variant TV3 and TV12	-	-	[25]
Cai et al., 2011	Blood immune cells	-	-	No difference	-	[26]
De Mena et al., 2013	Brain	-	-	No difference	-	[27]
Eryilmaz et al., 2017	Blood immune cells	-	-	Hypomethylation	-	[28]
Ding et al., 2016	Plasma	-	-	-	Decrease miR-181a	[29]
Xing et al., 2020	Brain	-	-	-	Decrease miR-218	[30]
Serafin et al., 2015	Plasma	-	-	-	Increase miR-103a-3p	[31]
PINK1						
Muqit et al., 2006	Brain	Increase ∆1-PINK1	-	-	-	[32]
Blackinton et al., 2007	Brain: SN	-	No difference	-	-	[33]
Navarro-Sanchez et al., 2018	Brain: SN	-	-	No difference	-	[34]
Fazeli et al., 2020/Dos Santos et al., 2018	PBMC/CSF	-	-	-	Decrease miR-27a	[35,36]
DJ1						
Kumaran et al., 2009	Brain	Decrease	Decrease	-	-	[37]
Tan et al., 2016	Blood immune cells	-	-	No difference	-	[38]
Chen et al., 2017	Plasma	-	-	-	Increase miR-4639-5p	[39]
GBA						
Murphy et al., 2014	Brain	Decrease	No difference	-	-	[40]
Moors et al., 2019	Brain	-	No difference	-	-	[41]
Eryilmaz et al., 2017	Blood immune cells	-	-	No difference	-	[28]

sPD: sporadic Parkinson’s disease; PBMC: peripheral blood mononuclear cell; CSF: cerebrospinal fluid; DA: dopaminergic; SN: *Substantia nigra*; controls: general population without PD.

## Data Availability

Not applicable.

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
