# Peer review of "Does the Expression and Epigenetics of Genes Involved in Monogenic Forms of Parkinson’s Disease Influence Sporadic Forms?"

_genes, 2022, doi:10.3390/genes13030479_

Round 1
Reviewer 1 Report
This review by Lanore et al. overviews the epigenetic mechanisms regulating the genes that are responsible for the development of sporadic Parkinson's disease (PD). They specifically discuss differences in the expression of relevant genes between sporadic human PD and controls, with additional considerations of histone modification, DNA methylation, and miRNA expression. The topic of this review is of great importance and interest, but there are several points listed below needed to be further revised before I can recommend its publication to Genes.
Major comments:
- In line 89-94, the authors talked about the repressive role of alpha-Syn to the gene encoding PGC-1alpha. However, I don’t see the logic of putting the paragraph in its current form here. The authors mentioned the decrease in PGC-1alpha would increase the “vulnerability to mitochondrial toxins”, but how does that relate to PD? I believe the discussion should be much more expanded.
- In line 140-149, I wonder what does the histone modification profiles look like at the SNCA locus? What about the relationship between other histone modifications than the H3K27 acetylation with SNCA expression, if there are any other histone marks existing in this region?
- I wonder if the authors can provide discussions on any previous evidence showing the relative weight of different factors including DNA methylation, histone modification, ncRNA, etc. in contributing to the regulation of SNCA expression, if there is any? How might these different factors interact with each other?
- In line 229-231, the authors mention that the epigenetic modifications, as well as microenvironmental changes, contribute to miR-205 dysregulation in cancer. How does this contribution happen? The authors need to expand the discussion.
- Similarly, in line 273-275, the authors mention that Post-translational modifications of Parkin and decrease in solute Parkin through sequestration in alpha-Syn aggregates are potential mechanisms for PD. More discussions are needed, from either the cited paper or opinions of the authors’ own.
- Similarly, the authors mention “PINK1 is not differentially expressed in PD patient brains”, I think more discussions on why this is observed should be provided?
- Similarly, in line 423-427, the authors mainly state the simple correlation and observations, but providing more mechanistic discussions or even reasonable conjecture would be valuable.
- I guess regarding the above four comments, the point is that I have a general impression that the current manuscript is more like a pile-up of previous findings but there are no deeper insights/discussions or penetrative interpretations/perspectives from the authors themselves. Opinions, even though may not be possible to be verified given the current efforts, are equally, or even more valuable than knowing what has been observed in previous experiments.
- Lastly, I wonder if it would be helpful to think through a different angle of the relationship between epigenetic modifications and PD disease development. Some very recent studies show that the epigenetic modifications are the main regulators of three-dimensional genome structure quantitatively (e.g. PMID: 29365171, PMID: 31181064) as well as its potential relationship with disease (PMID: 28964259, PMID: 31515496). I think it would be an interesting supplement to mention briefly from this structural perspective at the very end of the discussion section, which could possibly provide a new thinking pipeline of mechanistically how epigenetic modification connects with PD disease development.
Minor comments:
- The format of the text should be more organized. For example, in Table 1 “DNA methylation”, line 87, line 207 (and maybe other places) there should be a blank between the comma and the next word. In line 88, the citation format is chaotic. The authors should check other places in the manuscript.
- The figures are of low quality. And figure captions should be better written and more informative, especially Figure 1 and 3.
- There are two “Figure 2”. The third figure should be “Figure 3”.
Author Response
Thank you for these pertinent remarks, we have tried to complete as much as possible according to your comments.
- In line 89-94, the authors talked about the repressive role of alpha-Syn to the gene encoding PGC-1alpha. However, I don’t see the logic of putting the paragraph in its current form here. The authors mentioned the decrease in PGC-1alpha would increase the “vulnerability to mitochondrial toxins”, but how does that relate to PD? I believe the discussion should be much more expanded
- Indeed, we have moved the paragraph to the histone modification section. Rewording the implication of PGC1-a depletion: reported to lead to loss of dopamine neurons
- In line 140-149, I wonder what does the histone modification profiles look like at the SNCA locus? What about the relationship between other histone modifications than the H3K27 acetylation with SNCA expression, if there are any other histone marks existing in this region?
- We found a recent study on the profile of histone modification on SNCA that allowed us to complete our review. It turns out that this study additionally analysed the presence of H3K27 which was not different between parkinson and controls. We also added the difference observed on H3K4me3
- I wonder if the authors can provide discussions on any previous evidence showing the relative weight of different factors including DNA methylation, histone modification, ncRNA, etc. in contributing to the regulation of SNCA expression, if there is any? How might these different factors interact with each other?
- Unfortunately, no study has directly considered the different types of epigenetic modification. There are arguments in the literature for a correlation between H3K4 enrichment and DNA methylation, which is consistent with the observations made. Interest in performing mutlivariate analyses.
- In line 229-231, the authors mention that the epigenetic modifications, as well as microenvironmental changes, contribute to miR-205 dysregulation in cancer. How does this contribution happen? The authors need to expand the discussion.
- Indeed we did not go into detail on this, we added a discussion on the regulation of Mir-205 by hypoxia and inflammation.
- Similarly, in line 273-275, the authors mention that Post-translational modifications of Parkin and decrease in solute Parkin through sequestration in alpha-Syn aggregates are potential mechanisms for PD. More discussions are needed, from either the cited paper or opinions of the authors’ own.
- Although this is not directly the focus of the review, it is interesting. We have added a discussion on post-translational modifications of Parkin: sulphonation and S-nitrosylation: impact on E3 ligase activity but also on solubility, which could influence its aggregation and thus its availability
- Similarly, the authors mention “PINK1 is not differentially expressed in PD patient brains”, I think more discussions on why this is observed should be provided?
- After careful review, the study on which this observation was based was not accurate, and the interpretation of the results was unclear without direct comparison. We have therefore removed it from this review. No reliable information available on PINK1, however another study showed an increase in a ∆1-PINK1 degradation product. In addition we have added a discussion of the regulation of PINK1 expression by inflammation
- Similarly, in line 423-427, the authors mainly state the simple correlation and observations, but providing more mechanistic discussions or even reasonable conjecture would be valuable.
- We wanted to be factual about the mechanisms observed by giving elements so that readers could form an opinion. We went a little further in the hypotheses even if they remain conjectures. Proposed mechanisms: Inflammation and miR-27a / PINK1. Oxidative stress and MiR-494-3p and DJ-1
- I guess regarding the above four comments, the point is that I have a general impression that the current manuscript is more like a pile-up of previous findings but there are no deeper insights/discussions or penetrative interpretations/perspectives from the authors themselves. Opinions, even though may not be possible to be verified given the current efforts, are equally, or even more valuable than knowing what has been observed in previous experiments.
- See previous comment
- Lastly, I wonder if it would be helpful to think through a different angle of the relationship between epigenetic modifications and PD disease development. Some very recent studies show that the epigenetic modifications are the main regulators of three-dimensional genome structure quantitatively (e.g. PMID: 29365171, PMID: 31181064) as well as its potential relationship with disease (PMID: 28964259, PMID: 31515496). I think it would be an interesting supplement to mention briefly from this structural perspective at the very end of the discussion section, which could possibly provide a new thinking pipeline of mechanistically how epigenetic modification connects with PD disease development.
- Thank you for this interesting observation, indeed we did not include it in our review as it had not been explored in the context of Parkinson's disease. We have added this opening at the end of the article to open up new perspectives of analysis.
The text format has been rechecked.
Importing the figures into Word unfortunately reduces their quality, we have them in image format if needed.
The names of the figures have been changed accordingly.
Reviewer 2 Report
The authors review the genes involved in familial forms of Parkinson’s disease and question whether epigenetic dysregulation of these genes is involved in sporadic PD. Although there are several review articles on various epigenetic mechanisms in PD, including recent ones, this review article is focused on the regulation of genes linked with the familiar forms of PD. The ideas in the manuscript are relevant and up to date. It is well written and comprehensive.
Only few points to improve:
- Table 1 has too much text, not suitable for a table, and it lacks focus. I would be better if this table will be focused on PD, give example of epigenetic changes seen in PD, which type of PD and give references. The definitions of epigenetic factors can be given in the text rather than a table.
- for “human iPSC-derived neurons have been developed” add ref Kriks et al., Nature 2011
Author Response
Thanks for your feedback, we have removed the table and put the description of the epigenetic mechanisms involved back in text format. The description of the epigenetic mechanisms observed in sporadic Parkinson's disease is summarised in table 1.
We have also added this missing reference concerning the development of human iPSC-derived neurons
Other changes to the other reviewer's comments are for information:
we have moved the paragraph 89-94 to the histone modification section. Rewording the implication of PGC1-a depletion: reported to lead to loss of dopamine neurons
We found a recent study on the profile of histone modification on SNCA that allowed us to complete our review. It turns out that this study additionally analysed the presence of H3K27 which was not different between parkinson and controls. We also added the difference observed on H3K4me3
no study has directly considered the different types of epigenetic modification. There are arguments in the literature for a correlation between H3K4 enrichment and DNA methylation, which is consistent with the observations made. Interest in performing mutlivariate analyses.
we added a discussion on the regulation of Mir-205 by hypoxia and inflammation.
We have added a discussion on post-translational modifications of Parkin: sulphonation and S-nitrosylation: impact on E3 ligase activity but also on solubility, which could influence its aggregation and thus its availability
After careful review, the study on PINK1 expression was not accurate, and the interpretation of the results was unclear without direct comparison. We have therefore removed it from this review. No reliable information available on PINK1, however another study showed an increase in a ∆1-PINK1 degradation product. In addition we have added a discussion of the regulation of PINK1 expression by inflammation
We went a little further in the hypotheses even if they remain conjectures. Proposed mechanisms: Inflammation and miR-27a / PINK1. Oxidative stress and MiR-494-3p and DJ-1
We add a conclusion on regulatation of three-dimensional genome structure
Round 2
Reviewer 1 Report
The authors have addressed most of my concerns and the revised manuscript has largely improved. There is only one minor comment left.
- Regarding my previous comment on how histone modification profiles look like at the SNCA (comment #2), the authors added more discussions on H3K27 acetylation and H3K4me3, but I think it would be helpful to have an explicit figure on this so that one could appreciate the difference in histone modification difference more easily.
I would suggest publication after the final addressing of the above comment.
Author Response
Thank you for your feedback, indeed a figure is wise to realize the profile of the histone modification. We have added a figure using the Roadmap Epigenomics Database data